# A Firm-Level Investigation of Innovation in the Caribbean: A Comparison of Manufacturing and Service Firms

**Antonio Alleyne [1], Troy Lorde [2],\* and Quinn Weekes [3]**

[1]  Department of Accounting, Dongbei University of Economics and Finance, Dalian 116023, China;
   ar.alleyne@hotmail.com

[2]  Department of Economics, University of the West Indies, Bridgetown BB11000, Barbados

[3]  Central Bank of Barbados, Bridgetown BB11000, Barbados; quinn.weekes@centralbank.org.bb

\*  Correspondence: troy.lorde@cavehill.uwi.edu; Tel.: +246-417-4272

**Abstract:** A lack of growth remains a major concern for Caribbean countries. Private sector development has been identified as vital in addressing this problem. Innovation, a necessary condition for competitiveness, is a key channel through which the private sector can help to stimulate growth. An analysis of innovation at the firm level for Caribbean manufacturing and services sectors shows that patent rights, the level of domestic sales, collaboration for innovation purposes, innovation intensity (that is, the efficiency with which innovation funds are managed), availability of technology, knowledge about new market trends, domestic sales, and the size of the workforce are critical to the innovation process in both sectors. Several differences also exist. Innovative service firms are older, in contrast to manufacturing firms, which tend to be younger; foreign ownership is key for service firms; and both types of firms face different obstacles to innovation. Policymakers should tailor policies that take such differences into account.

**Keywords:** innovation; firm level; Caribbean; manufacturing; services

**JEL Classification:** C55; O14; O30

## 1. Introduction

Economic growth remains a serious challenge for Caribbean countries. While economic performance among countries varies, they each share a daunting outlook. Several have had little to no growth in recent years. The last significant growth periods occurred when Caribbean countries still enjoyed preferential arrangements with Europe for bananas and sugar, which have since eroded, and in the early 21st century from tourism, which has stagnated due to the maturity of the product in several states and the emergence of other tourism destinations. Further, regional countries are severely indebted and have very limited fiscal space to promote growth from public investment. Faced with the current circumstances, prospects for economic growth in the Caribbean will depend on the ability of the private sector to increase its productivity and competitiveness. Indeed, the European Commission notes that, "as a driver of inclusive growth and job creation, responsible for 84% of GDP and 90% of jobs in developing countries, the private sector is ideally placed to improve the lives of the poor and deliver on the promise of sustainable and socially inclusive economic development", and that "private sector development plays a key role in creating economic growth, employment and improved living

conditions".[1] Private sector performance in the Caribbean, though, has been lackluster. The sector is beset with key challenges, for example, operating in small domestic markets vulnerable to external shocks, coping with persistent structural problems in the business environment, and their own lack of enterprise.

A key channel through which the private sector can enhance its productivity and competitiveness is innovation. Innovation, the successful commercialization of novel ideas, including products, services, processes and business models, is a critical component of economic growth. Indeed, recent studies show that innovation is a major source of growth and a necessary condition for competitiveness (Crespi and Zúñiga 2012; Mahmud and Ahmed 2011). Growth from innovation occurs in two complementary ways: by introducing new or improved products that tap into existing or latent demand in the market, thereby creating additional value for firms and consumers; and, by increasing the productivity of firms employing such innovations (WEF World Economic Forum).

Little, however, is known about innovation in the Caribbean. Only five countries appear in the rankings of The Global Innovation Index, Barbados, the Dominican Republic, Guyana, Jamaica and Trinidad and Tobago (Table 1).

**Table 1.** Innovation Performance: Global Innovation Index Rankings and Scores.

|  | **2013** | **2014** | **2015** | **2016** |
|---|---|---|---|---|
| Barbados | 47 (40.5) | 41 (40.8) | 37 (42.5) | – |
| Dominican Republic | 79 (33.3) | 83 (32.3) | 89 (30.6) | – |
| Guyana | 78 (34.4) | 80 (32.5) | – | – |
| Jamaica | 82 (32.9) | 82 (32.4) | 96 (29.9) | 89 (29) |
| Trinidad and Tobago | 81 (33.2) | 90 (31.6) | 80 (32.2) | – |
| No. of countries ranked in GII | 142 | 143 | 141 | 128 |

Source: Global Innovation Index (GII) Online Database. Note: Values not in parentheses are global rankings while values in parentheses are GII indices.

Barbados is ranked in the top one-third of all countries that appeared in the GII between 2013 and 2015, and showed improvement both relatively (in terms of rankings) and absolutely (in terms of the index score) in innovation over time. Jamaica ranked in the bottom half of the GII for 2013–2014 and the bottom third for 2015–2016. While its ranking fluctuated, the trend of Jamaica's index score indicates that the level of innovation declined in the country. The Dominican Republic, Guyana and Trinidad and Tobago are ranked in the bottom half of the GII from 2013 to 2015. With the exception of Trinidad and Tobago whose scores fluctuated, GII scores declined, suggesting that innovation did not improve over the period. The Global Competitive Index (GCI) also provides some additional insight into the state of innovation in the region. The Caribbean[2] is ranked lowest in all pillars of the GCI, with the exception of the macroeconomic environment, between 2005 and 2015 (Figure 1).

As it relates specifically to the innovation pillar, the Caribbean's innovation level lies significantly below other regions: the Caribbean is measured at 2.8; ASEAN, 3.6; EU28, 4.1; and the USA, 5.5, virtually double that of the Caribbean. At a country level, GCI scores for Caribbean countries range

---

[1]  Private Sector Development. Available online: http://ec.europa.eu/europeaid/node/679_hr (accessed on 29 August 2017).
[2]  The GCI innovation score for the Caribbean is based on the six members that are captured in the WEF reports between 2005 and 2015: Barbados, Guyana, Haiti, Jamaica, Suriname and Trinidad and Tobago.

from 2 (Haiti) to 3.6 (Barbados), compared to higher levels of innovation recorded for other countries (Figure 2).

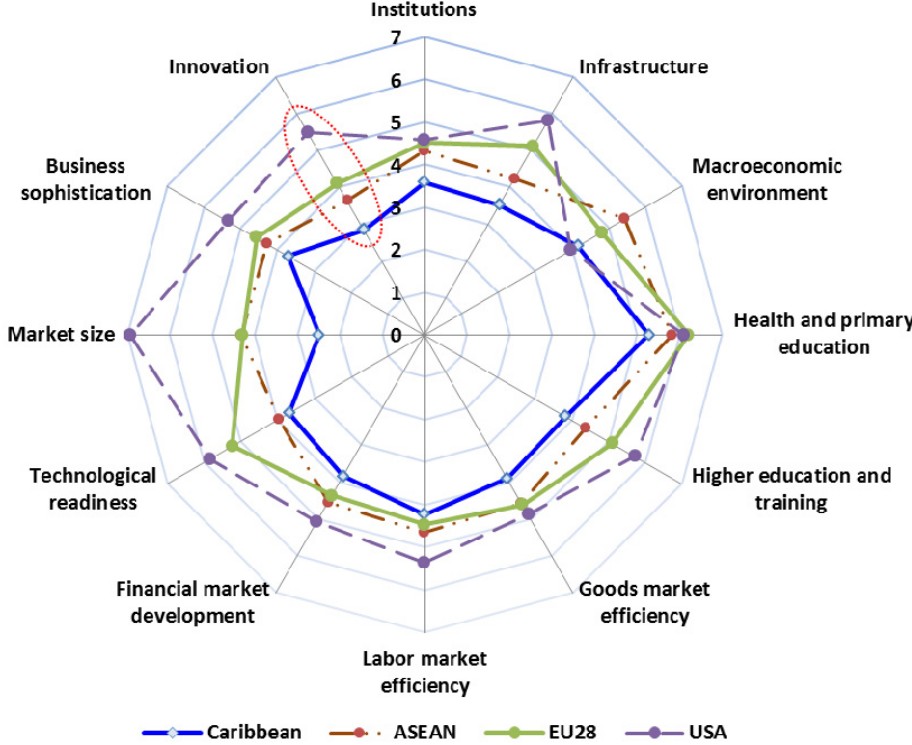

**Figure 1.** Comparison of Innovation: Caribbean vs. ASEAN, EU28 and USA. Source: Authors' calculations based on World Economic Forum Global Competitiveness Index for 2005–2015.

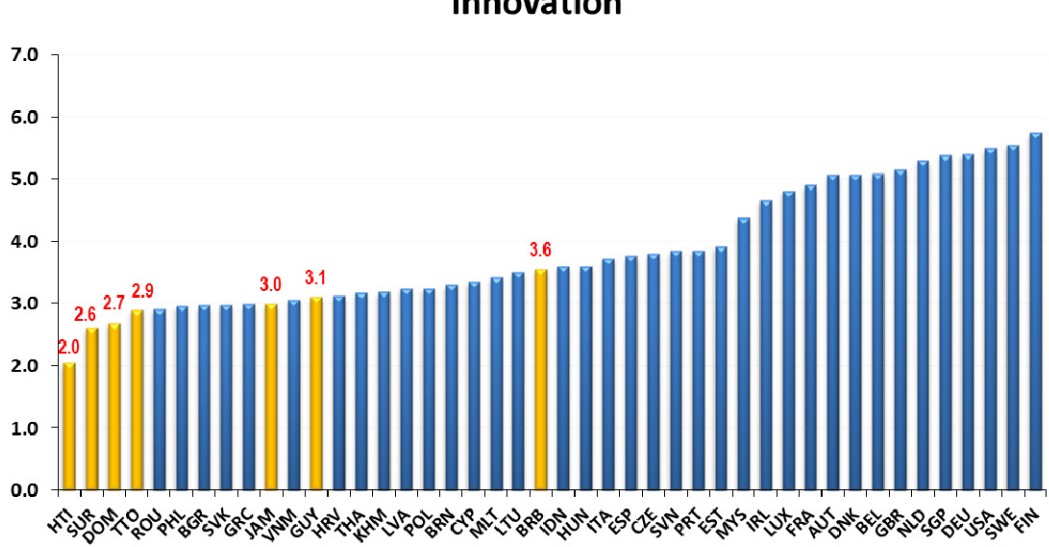

**Figure 2.** Global Comparison of Innovation in Individual Caribbean Countries. Source: Authors' calculations based on World Economic Forum Global Competitiveness Index for 2005–2015.

The scant available evidence strongly suggests that the Caribbean lags behind other regions and urgently needs to improve its level and rate of innovativeness given the constantly shifting external context for economies and the increasingly competitive global environment. It is not known with any degree of certainty the underlying causes for the poor innovative performance of the region. Possible

reasons include: low R&D intensity (0.82% of GDP in 2013 in Latin America and the Caribbean region vs. OECD and global averages of 2.43% and 2.13% respectively);[3] low private sector participation in innovation efforts (Navarro et al. 2016); sub-optimal funding mechanisms for innovation purposes; a shortage of human capital for innovation (509 persons per million in R&D in Latin America and the Caribbean in 2010 vs. OECD and global averages of 3344 and 1282 per million);[4] a weak innovation climate (Navarro et al. 2016); and weak institutional capacity (Navarro et al. 2016).

It is against this background that the current paper conducts an analysis of innovation in the Caribbean at the firm level. The extent to which little is known, particularly at the firm level, has implications for regional policymakers who lack information on which to craft suitable incentives for innovative activity. Findings could assist policymakers, firms and other stakeholders seeking to increase competitiveness, as the right innovative environment requires appropriate policies and practices to be in place (Chen and Huang 2009). More generally, findings could provide insight into innovation in small states, which are underrepresented in the existing literature.

Another area of departure is that this study investigates innovation in both the manufacturing and services sectors. Most studies of innovation continue to focus on manufacturing. Due to the economic importance of the service sector, greater attention on innovation in services is appropriate. Over 60% of the value-added in global GDP in 2015 was produced in the service sector,[5] yet we understand little about the sector's underlying drivers. Further, the boundary between services and manufacturing is getting blurrier (Christensen and Drejer 2007) as a significant degree of innovation in the manufacturing sector involves service activities. Thus, analyses of innovation in services may shed some light on aspects of innovation in the manufacturing sector.

The rest of the paper is organized as follows. Section 2 presents a concise review of the relevant literature. Section 3 describes the methods and data used. This is followed by a presentation of the empirical findings in Section 4. The final section provides concluding remarks and discusses the implications of the findings.

## 2. Review of Literature

Multiple definitions of innovation are found in the literature. The most common refers to the introduction of a new product or process within the last three years (Mahmud and Ahmed 2011). Recently, Edison et al. (2013) note that innovation is the production or adoption, assimilation, and exploitation of a value-added novelty in economic and social spheres; renewal and enlargement of products, services, and markets; development of new methods of production; and establishment of new management systems. Murphy and Nordfors (2006) suggest that individuals invent but innovation must be undertaken in communities. The authors define innovation as the introduction of a new concept, not always based on advanced technologies or complicated business models. Murphy & Nordfors emphasize the importance of innovation to the competitiveness of both developing and developed economies. Becheikh et al. (2006) find that 81% of innovation studies investigated process, product, or both types of innovative activity. HM Treasury (2006) sees innovation as a key catalyst for increasing economic productivity, driving enterprise, creating new products and markets, and improving efficiency, delivering benefits to firms, customers and society in general. Further, innovation is about delivering change and the successful exploitation of new ideas to provide economic or social value. Innovation is therefore both a process and a product (Edison et al. 2013).

At first blush, these definitions appear inapt when considered within the context of developing countries, which lag behind developed countries in R&D investment and the introduction of new products and processes. For the former, R&D entails increasing absorptive capacity of foreign R&D

---

3   Statistics collected from World Bank *World Development Indicators* online databank: https://data.worldbank.org/data-catalog/world-development-indicators.
4   Ibid.
5   Ibid.

(Cohen and Levinthal 1989), and innovation means efficient adaptation and harnessing of externally developed technology and knowledge (El Elj and Abassi 2014). Indeed, firms in developing countries are more capacitated to absorbing technologies rather than creating their own (Bogliacino et al. 2012). Yet, while innovation activities in developing countries often occur at the margins and are based on implementing technologies and products already available elsewhere, they can still allow for the development of a comparative advantage. Thus, understanding the main factors that drive innovation is also important for developing countries (Goedhuys 2007; Dabla-Norris et al. 2012).

There is a substantial literature on the role that innovation plays in firm performance. The strand which can be traced to Schumpeter (1934, 1942)'s seminal works, argues that a firm's decision to innovate is a function of firm characteristics such as size, industry features, market concentration and sector sophistication (Cohen 1995). More recent studies have identified other factors that can explain the propensity to innovate. These include, technological capability, managerial ability, human capital, adoption and mastery of external technologies, among others.

Firm size has been identified as one of the foremost traditional determinants of innovation in developed countries (Cohen 1995). This relationship has been examined in developing countries, and has also been found significant (Pamukcu 2003; Rahmouni et al. 2010). Larger firms benefit from economies of scale and have greater access to financial resources. This permits them to acquire new technologies by licensing and patented innovations, which are important sources of technology access for developing countries (Almeida and Fernandes 2008). There is also a reputational and experience effect of larger firms, which facilitates their cooperation with research centers and offshore companies (El Elj and Abassi 2014). The literature also shows that younger firms are more inclined to innovate than older firms (Ayyagari et al. 2011) as older forms are more risk-averse.

Integration into global markets has drawn recent attention for its impact on innovation. Theoretically, trade openness should encourage firms to innovate and enhance their competitiveness at the global level. Some evidence for developing countries has shown that exporting firms innovate more than non-exporting firms (Almeida and Fernandes 2008). Competition in both local and foreign markets compel exporting firms to innovate to maintain and enhance their competitiveness (Ayyagari et al. 2011). Exporting firms may also have easier access to foreign technology (El Elj and Abassi 2014).

As indicated previously, developing countries are more oriented towards absorption of technology as opposed to creating their own. This capacity is determined in part by technological, managerial and human capital skills. Pamukcu (2003) and Almeida and Fernandes (2008) find that the quality of human resources (measured by the experience and educational attainment of managers) enhances innovation. However, qualifications are not always equivalent to skills, and may be a misallocation of human resources in the innovation process (El Elj 2012). The licensing of technology improves the likelihood of innovation, as it represents a key source of external technology acquisition (Almeida and Fernandes 2008).

Several differences exist between service and manufacturing innovation. The essential difference is that most services are intangible and often co-produced with clients (Gallouj and Weinstein 1997). Development of service projects is usually undertaken by informal ad hoc committees or project teams, as opposed to by full-time R&D departments (Leiponen 2005; Miles 2007; Sundbo 1997). In-house training has been argued to compensate for formal R&D activities in some business service firms (Leiponen 2005). Additionally, despite the importance of technology to the provision of services, it is not a necessary condition for service innovation, which can take place even in the absence of technical factors (Zhang and Zhang 2007).

Further differences between services and manufacturing include the sources of knowledge for innovation and the use of intellectual property rights to protect the returns on investments in innovation. Arundel et al. (2007) note that universities and research institutes are less valued as sources of knowledge for service innovators relative to manufacturing innovators. In addition, both Arundel et al. (2007) and Tether and Massini (2007) find that service innovators utilize formal intellectual property rights much less intensively than product innovators do. Patents are the

second-most important method of protection for manufacturing firms but only the sixth-most important for services (Tether and Massini 2007).

Research also indicates that the types of innovation investment by service and manufacturing firms differ greatly. Crespi et al. (2014) further suggest that manufacturing firms invest more in machinery acquisition, while service firms focus on engineering and industrial design, disembodied technology, training, and marketing activities. The latter can perhaps explain the greater heterogeneity in innovation activities in services compared to manufacturing (Sirilli and Evangelista 1998).

There is little consensus about the indicators used to measure innovation performance. Cordero (1990) finds that performance evaluation focuses mainly on resources and outputs—R&D expenditure, speed to market, market-share, and number of new products—and ignores the processes in between. Wallin Johanna et al. (2011) argue that inadequacies also exist when primary focus rests on financial metrics, indicating that a firm's past performance is often a poor predictor of future success. Moreover, that a successful business does not necessarily imply a high innovation capacity, as poor innovation practices can lead to success, whereas good innovation practices may result in unwanted outcomes. Werner and Souder (1997) discuss the use of objective versus subjective, and quantitative versus qualitative metrics; a combination of methods would reduce biases, take advantage of multiple dimensions of excellence and provide built-in checks and balances to capture the full range of the R&D processes. Olsson (2008) offers that measuring the effects of innovation are ultimately based on the author's definition, noting that it is necessary for individual organizations to determine what is important to measure with regard to their specific circumstances and goals.

The business environment has an impact on the manner in which innovation is undertaken. In both large and small firms, innovation is preferably introduced by way of incrementally innovative products to manage the potential risk caused by challenging financial conditions, (Gamal et al. 2011). This approach has its drawbacks, as there is a tendency to reduce the potential for higher rewards (Gamal et al. 2011). The policy mix is also a significant barrier to the innovation cycle, particularly for developing nations. Rubalcaba (2013) stresses the need for effective policy to promote innovative initiatives, but acknowledges the difficulty in deriving appropriate policies for services, due to the unique complexity of determining service innovation. Service innovation policies can therefore be seen as experimental, which requires intermittent adjustments for maximum impact. Strong patent rights may complement competition-increasing product market reforms in fostering innovation. Similarly, product market reform may enhance innovative investments in manufacturing industries where strong patent rights exist (Aghion et al. 2015). Additionally, better patent protection prolongs the period over which a firm successfully escapes competition by innovating. Thus, innovation requires long-term commitment (Brenner 1994).

Scant evidence exists of the characteristics and determinants of innovation in the Caribbean. This is mainly due to a dearth of data. Rubalcaba (2013) shows that apart from telecommunications, the services sector in Latin America and the Caribbean (LAC) is characterized by low productivity resulting in low growth. Rubalcaba thus argues that due to the transformative power of service innovation in any economic activity, an essential factor for increasing the potential for growth and productivity in LAC is service innovation. The promotion of innovation in LAC is particularly important in contexts where innovative efforts are hampered by the weak linkages that characterize their national innovation systems (Crespi and Zúñiga 2012).

## 3. Research Methods

Models derived from the work of Crépon et al. (1998) have been very successful in identifying the determinants of innovation (Crespi and Zúñiga 2012). The strand of models is structured as follows: (i) firm decides to engage in innovation activities; (ii) firm decides the intensity of the investment in innovation activities; (iii) the knowledge or innovation production function (output) as a consequence of the innovation investments (inputs); and (iv) the impact on product or productivity of the knowledge produced along with other inputs.

An issue may arise when seeking to employ CDM models for the services sector. According to Organisation for Economic Co-Operation and Development (2009), the method relies on R&D as a proxy identifier for firm innovation. Service firms may find it difficult to adequately account for specific R&D expenditures. R&D can be carried out in formal R&D departments or informally in facilities where R&D is not the main activity. Identifying informal R&D may be difficult or costly in some businesses. Because service innovations are customer related, R&D activities are often integrated. Stand-alone R&D departments are unlikely where such activities can be readily quantified by existing accounting systems. It is also common for an employee to multitask with their primary duties and contribute to innovative processes. This makes quantifying R&D difficult, because fractions of expenditures based on existing accounting systems need to be assigned as R&D (RTI International 2005).

### 3.1. The Model

The model draws on the seminal work of Heckman (1978), along with the later modification by De Fuentes et al. (2015). The three-stage model explores the main factors underpinning a firm's decision to invest in innovation, and the intensity of the investment in innovation: (i) the first stage should correct for selection bias as not all firms engage in innovation; (ii) the second stage focuses on the innovation output, measured as product and process innovations; and, (iii) the third stage concentrates on the performance impact of innovation (on productivity) in firms.

The first stage of Heckman's model, comprised of two equations, focuses on the problem of selectivity bias. A firm's decision to innovate is jointly determined by a firm's choice of the ownership, that is, a series of firm-specific characteristics. The first equation, the selection equation, identifies the main elements for innovating:

$$\text{inf}_i = \beta X_i + \delta Q_i + \varepsilon_i \ \text{ else } \ \text{inf}_i = 0 \rightarrow \varepsilon_i \sim N(0,1) \tag{1}$$

Dependent variable ($\text{inf}_i$) is a binary variable which takes a value of 1 if the firm performs at least one type of innovation activity; $X_i$ is a vector of firm-specific characteristics such as size and age; $Q_i$ is a vector of instruments including ownership, accounts for exports, patents, the use of public funds to innovate; and $\varepsilon_1$ is a disturbance term.

In the second equation, the determinants of innovation intensity are investigated:

$$\ln \text{inf}_{2i} = \beta X_i + \delta L_i + \varepsilon_i \ \ \varepsilon_i \sim N(0,\sigma) \tag{2}$$

Ininf$_{2i}$ is the log of the firm's innovation investment per employee/innovation intensity. The vector of explanatory variables ($L_i$) accounts for exports, ownership, patents, the use of public funds to innovate, openness strategy, sources of information and factors that hamper the innovative or potentially innovative firms.[6] The likelihood function corresponding to above events is a bivariate probit model, which will be estimated via Heckman's two-step consistent estimator under robust conditions.[7] Consequently, the predicted probability of Equations (1) and (2), $\tilde{\pi}^a$ and $\tilde{\pi}^b$ respectively, that $\text{inf}_i = 1$ and Ininf$_{2i} > 0$ are:

---

[6]　In the literature, impediments to the innovation process have been categorised under four main classifications; i.e., cost related, institutional/organisational practices, market forces, and knowledge.

[7]　Despite arguments that likelihood estimation approaches are theoretically superior, Heckman (1978) suggested an unassuming two-step method for taking care of endogeneity, which works under noted conditions. This method has been applied to probit response models, recently by various researchers. According to Freedman and Sekhon (2010), significant numerical challenges are faced when trying to maximise the bi-probit likelihood function, as required under this study, even if the number of covariates is small. Heckman's test under probit model assumptions can be suggested as a useful diagnostic.

$$\begin{cases} \widetilde{\pi}^\alpha = \widetilde{\Pr}(\inf_i = 1) = F(\widetilde{Z_I}^\alpha = \Pr(\varepsilon_1 < \widetilde{Z_i}^\alpha), \\ \widetilde{\pi}^b = \widetilde{\Pr}(\ln \inf_{2i}{}^* = 1) = \widetilde{\Pr}(\ln \inf_{2i} > 0) = F(\widetilde{Z_i}^b = \Pr(\varepsilon_1 < \widetilde{Z_i}^b) \\ Z_i{}^\alpha = Z_i{}^b = \alpha_0 + \beta\widetilde{X_i} + \delta\widetilde{L_i} \end{cases} \quad (3)$$

In the second stage of the model, the innovation expenditure function, a firm's innovation output, is measured by the introduction of product or process innovations ($innov_i$). Here, the independent variables include the predicted expenditure probability from Equation (2) [$\Pr(\text{Ininf}_{2i})$]. This equation is as follows:

$$immov_i = \beta X_i + \widetilde{\Pr}(\ln \inf_{2i}) + \delta \text{L}_i + \varepsilon_1 \ \text{else} \ innov_i = 0 \quad (4)$$

The final stage of this analysis captures the impact of innovation on a firm's performance per employee expressed in logarithms. The independent variables include the predictor from Equation (4) [$\Pr(innov_i)$]:

$$\ln prod_i = \beta X_i + \widetilde{\Pr}(innov_i) + \delta \text{L}_i + \varepsilon_i \quad (5)$$

The model is estimated separately for manufacturing and service firms.

### 3.2. Variables

The variables used in this analysis are based on consideration of previous work by Crespi and Zúñiga (2012), Crespi et al. (2014), and De Fuentes et al. (2015). Table A1 in the Appendix A contains the variable names, their descriptions and the equations in which they appear for the model.

### 3.3. Data

Observations are taken from the Productivity, Technology and Innovation (PROTEqIN) survey of firms in the Caribbean by *Compete Caribbean*[8] in 2013–2014 (Table 2). The dataset is comprised of 660 manufacturing firms and 1268 service firms, all with five or more employees.

Retail establishments (23.7%) and Hotel and Restaurants (17.2%) respectively, both from the services sector, comprise the single largest shares of the sample. Food manufacturers comprise the largest group of firms from the manufacturing sector (10%).

Shareholding companies are the most common form of establishment in the manufacturing sector (40.9%), in contrast with the services sector where sole proprietorships dominate (39.6%) (Table 3). Firms from each sector are majority domestic-owned (88%). Government ownership is marginal, at around 0.6% for both sectors.

Sales by Caribbean firms are home-biased; 88.3% of manufacturing output and 93.1% of service output is sold to domestic markets. Only 22.9% of firms possess internationally recognized certification—25.3% of manufacturing firms and 21.6% of service firms. On average, the value of foreign inputs is 40.5% of sales (36.2% of manufacturing and 43.0% of service firms). The main export market for most firms is the USA.

An inadequately educated labor force is the most significant obstacle to success cited by manufacturing firms (59.1%), while for service firms, access to finance poses the largest problem (88.1%). The cost of finance, the degree of taxation, political interference, forms of crime, and competition from the informal sector, among others, are also highlighted as key environmental challenges to the operational survival of firms.

---

8　Compete Caribbean is a program, jointly funded by the Inter-American Development Bank (IDB), the United Kingdom Department for International Development (DFID) and the Government of Canada, that provides technical assistance grants and investment funding to support productive development policies, business climate reforms, clustering initiatives and Small and Medium Size Enterprise (SME) development activities in the Caribbean.

**Table 2.** Structure of Manufacturing and Service Firms in CARICOM Countries.

| Firm Type | ATG | BRB | BLZ | DOM | GRD | GUY | JAM | SLU | SKN | SVG | SUR | BAH | TTO | Freq. | % |
|---|---|---|---|---|---|---|---|---|---|---|---|---|---|---|---|
| **Manufacturing** | | | | | | | | | | | | | | | |
| Food | 8 | 14 | 22 | 13 | 11 | 17 | 26 | 2 | 5 | 21 | 19 | 16 | 22 | 196 | 10 |
| Textiles | 0 | 3 | 0 | 0 | 0 | 0 | 0 | 0 | 0 | 1 | 2 | 0 | 0 | 6 | 0.3 |
| Garments | 1 | 5 | 1 | 0 | 1 | 3 | 10 | 4 | 3 | 2 | 2 | 1 | 8 | 41 | 2.1 |
| Chemicals | 2 | 5 | 5 | 6 | 1 | 3 | 4 | 3 | 0 | 0 | 1 | 6 | 15 | 51 | 2.6 |
| Plastics and rubber | 0 | 3 | 1 | 0 | 0 | 0 | 3 | 0 | 2 | 1 | 3 | 0 | 5 | 18 | 0.9 |
| Non-metallic mineral products | 1 | 5 | 3 | 0 | 1 | 0 | 7 | 14 | 1 | 5 | 3 | 3 | 6 | 49 | 2.5 |
| Basic metals | 0 | 0 | 0 | 0 | 0 | 0 | 0 | 19 | 0 | 1 | 0 | 0 | 7 | 27 | 1.4 |
| Fabricated metal products | 2 | 2 | 1 | 1 | 1 | 4 | 2 | 2 | 3 | 1 | 3 | 2 | 11 | 35 | 1.8 |
| Machinery and equipment | 8 | 0 | 1 | 2 | 4 | 1 | 2 | 9 | 6 | 4 | 1 | 4 | 3 | 45 | 2.3 |
| Electronics | 3 | 0 | 0 | 0 | 0 | 0 | 1 | 3 | 6 | 0 | 0 | 1 | 4 | 18 | 0.9 |
| Other manufacturing | 1 | 17 | 23 | 0 | 2 | 15 | 40 | 1 | 2 | 6 | 32 | 0 | 35 | 174 | 8.9 |
| **Services** | | | | | | | | | | | | | | | |
| Construction | 9 | 6 | 3 | 10 | 16 | 3 | 7 | 2 | 16 | 8 | 7 | 23 | 26 | 136 | 6.9 |
| Services of motor vehicles | 7 | 1 | 3 | 0 | 11 | 4 | 21 | 0 | 4 | 8 | 2 | 3 | 14 | 78 | 4 |
| Wholesale | 1 | 7 | 2 | 7 | 4 | 13 | 12 | 4 | 7 | 3 | 6 | 3 | 26 | 95 | 4.8 |
| Retail | 38 | 9 | 18 | 20 | 35 | 37 | 71 | 28 | 30 | 38 | 14 | 21 | 107 | 466 | 23.7 |
| Hotel and restaurants | 35 | 32 | 31 | 39 | 34 | 10 | 18 | 32 | 27 | 17 | 8 | 30 | 26 | 339 | 17.2 |
| Transport | 13 | 8 | 8 | 23 | 7 | 3 | 13 | 5 | 12 | 15 | 16 | 10 | 21 | 154 | 7.8 |
| Information technology | 2 | 6 | 0 | 5 | 1 | 7 | 5 | 0 | 1 | 2 | 1 | 4 | 4 | 38 | 1.9 |

Source: PROTEqIN. Note: Atg = Antigua & Barbuda; BRB = Barbados; BLZ = Belize; DOM = Dominica; GRD = Grenada; GUY = Guyana; JAM = Jamaica; SLU = St. Lucia; SKN = St. Kitts & Nevis; SVG = St. Vincent & the Grenadines; SUR = Suriname; BAH = the Bahamas; and, TTO = Trinidad & Tobago.

**Table 3.** Characteristics of CARICOM Firms.

| Indicator | 2013/14 | | Indicator | 2013/14 | |
|---|---|---|---|---|---|
| **Labor and Skills** | **Mnu** | **Srv** | **Financing** | **Mnu** | **Srv** |
| Number of permanent full-time workers | 63.0 | 52.0 | Currently have a line of credit or loan from a financial institution | 42.6 | 37.9 |
| Number of temporary workers | 6.0 | 4.0 | Proportion of working capital financed internally (%) | 58.4 | 59.6 |
| Percent of firms offering formal training (%) | 56.5 | 56.4 | Proportion of working capital financed by banks (%) | 16.5 | 15.0 |
| Percent of firms identifying an inadequately educated workforce as a major constraint (%) | 59.1 | 86.4 | Proportion of working capital financed by supplier credit (%) | 16.7 | 18.4 |
| Percent of firms identifying labor regulations as a major constraint (%) | 36.7 | 17.3 | Proportion of working capital financed by Government (%) | 3.4 | 2.9 |
| **Legal Status** | | | **Business Environment Obstacles** | | |
| Shareholding company (%) | 40.9 | 34.0 | Tax rates (%) | 55.0 | 79.5 |
| Sole proprietorship (%) | 30.8 | 39.6 | Access to Finance (%) | 55.2 | 88.1 |
| Partnership (including limited liability (%) | 14.1 | 12.3 | Cost of Finance (%) | 55.9 | 79.1 |
| Limited partnership (%) | 13.6 | 13.9 | Macroeconomic Conditions (%) | 48.9 | 77.9 |
| Other (%) | 0.6 | 0.2 | Customs and Trade Regulations (%) | 46.7 | 79.7 |
| Proportion of private domestic ownership in a firm (%) | 88.1 | 88.0 | Political Environment (%) | 42.7 | 71.0 |
| Proportion of private foreign ownership in a firm (%) | 11.0 | 11.2 | Inadequately Educated Workforce (%) | 59.1 | 86.4 |
| Proportion of government/state ownership in a firm (%) | 0.6 | 0.6 | Electricity (%) | 48.8 | 76.3 |
| **Gender Composition of Management** | | | Practices of the Informal Sector (%) | 49.4 | 79.9 |
| All men (%) | 23.8 | 22.1 | Tax Administration (%) | 45.3 | 75.8 |
| Predominantly men (%) | 43.3 | 40.0 | Transportation (%) | 37.7 | 66.4 |
| Equally men and women (%) | 17.4 | 17.8 | Crime, Theft and Disorder (%) | 50.6 | 85.4 |
| Predominantly women (%) | 10.0 | 12.8 | **Business Strategy—Goals for past 2 years.** | | |
| All women (%) | 5.2 | 7.4 | To obtain quality certification (%) | 21.4 | 22.1 |
| **Sales, Foreign Trade Competition** | | | To support innovation (%) | 27.6 | 11.7 |
| Portion of firms with internationally Recognized Quality Certification (%) | 25.3 | 21.6 | To promote exports (%) | 17.3 | 15.2 |
| National Sales (%) | 88.3 | 93.1 | To improve quality of good or services (%) | 28.6 | 8.1 |
| Direct Exports (%) | 11.7 | 6.9 | To develop new foreign markets (%) | 11.8 | 12.7 |
| Primary Destination for Direct Exports (%) | USA | USA | To reduce energy consumption (%) | 30.3 | 27.8 |
| Material Inputs or supplies of foreign origin (% of sales) | 36.2 | 43.0 | To increase the number of goods or services offered (%) | 31.8 | 17.7 |

Note: **Mnu** = manufacturing firms and **Srv** = service firms.

## 4. Results

Results from estimation of the model are presented in Table 4 (manufacturing) and 5 (services). The analysis captures the main factors determining innovation activities within the manufacturing and services sectors, the estimated level of innovation expenditure of firms, and the average contribution of innovation to firms' operational performance.

### 4.1. Probability of Innovation and Innovation Intensity

Innovation activity in manufacturing and services firms is analyzed using two indicators, the probability of and intensity of innovation per employee (columns 2 and 3 of Tables 4 and 5).

**Table 4.** Innovation in CARICOM Manufacturing Firms.

| Independent Variables | Probability of Innovation Per Employee (*d_INF*—Equation (1)) | Innovation Intensity Per Employee (*lnINEM*—Equation (2)) | Innovation Output (*d_INNOV*—Equation (4)) | Innovation Productivity (*lnPROD*—Equation (5)) |
|---|---|---|---|---|
| *lnLABOR11* | | | | 0.607 *** (0.127) |
| *lnDSALES* | | 0.222 *** (0.05) | 0.748 *** (0.196) | |
| *lnESALES* | | | | |
| *d_O_STRATEGY* | | −1.553 *** (0.258) | | |
| *d_TECH_BASED* | | | 0.346 * (0.18) | |
| *d_HFDI* | | | | 0.427 ** (0.217) |
| *d_PAT_FILED* | 12.02 *** (0.027) | | | |
| *lnLABOR12* | | | 0.664 *** (0.119) | |
| *lnINEM_hat* | | | −3.419 *** (0.87) | 1.218 *** (0.2) |
| *d_FCULTURE_OBS* | | | | |
| *d_TECH_INFO_OBS* | | | −0.168 ** (0.084) | |
| *d_MK_INFO_OBS* | | | −0.202 *** (0.077) | |
| *d_FRM_OPEN_OBS* | | | | |
| *d_OTHER_OBS* | | | 0.029 *** (0.005) | |
| *d_INNOV(est.)* | | | | 2.15 *** (0.385) |
| *lnFIRM_AGE13* | | | | −0.137 *** (0.077) |
| *d_FINANCE_COST* | | | | 0.141 ** (0.056) |
| *d_BUS_LICENCE* | | | | 0.114 ** (0.048) |
| *d_ELECTRICITY* | | | | |
| *d_TAX* | | | | |
| *d_CRIME* | | | | |
| *_cons* | −6.142 *** (0.057) | 5.742 *** (0.793) | 18.83 *** (4.980) | |
| Observations | 537 | 537 | 591 | 591 |

Notes: Standard errors are in parentheses. ***, **, and * indicate significance at 1%, 5% and 10% respectively. Insignificant coefficients are excluded from the table of results.

**Table 5.** Innovation in CARICOM Service Firms.

| Independent Variables | Probability of Innovation Per Employee (*d_INF*—Equation (1)) | Innovation Intensity Per Employee (*lnINEM*—Equation (2)) | Innovation Output (*d_INNOV*—Equation (4)) | Innovation Productivity (*lnPROD*—Equation (5)) |
|---|---|---|---|---|
| *lnLABOR11* | 0.028 * (0.016) | −0.383 ** (0.16) | | |
| *lnDSALES* | −0.011 *** (0.004) | 0.331 *** (0.105) | 1.057 *** (0.403) | |
| *lnESALES* | | 0.072 *** (0.025) | 0.24 *** (0.087) | |
| *d_O_STRATEGY* | | −1.765 *** (0.484) | | |
| *d_TECH_BASED* | | 0.815 * (0.434) | 2.472 *** (0.989) | −0.720 *** (0.139) |
| *d_HFDI* | 0.069 * (0.041) | | | |
| *d_PAT_FILED* | 12.000 *** (2.38) | | | |
| *lnLABOR12* | | | −1.063 *** (0.47) | 0.710 *** (0.036) |
| *lnINEM_hat* | | | −3.184 *** (1.22) | 1.207 *** (0.108) |
| *d_FCULTURE_OBS* | | | −0.128 * (0.02) | |
| *d_TECH_INFO_OBS* | | | −0.127 * (0.067) | |
| *d_MK_INFO_OBS* | | | −0.13 * (0.0691) | |
| *d_FRM_OPEN_OBS* | | | −0.177 *** (0.063) | |
| *d_OTHER_OBS* | | | 0.031 *** (0.003) | |
| *d_INNOV(est.)* | | | | 2.68 *** (0.196) |
| *lnFIRM_AGE13* | | | | 0.156 *** (0.093) |
| *d_FINANCE_COST;* | | | | |
| *d_BUS_LICENCE* | | | | |
| *d_ELECTRICITY* | | | | 0.093 *** (0.033) |
| *d_TAX* | | | | 0.089 *** (0.034) |
| *d_CRIME* | | | | 0.124 *** (0.036) |
| *_cons* | −6.223 *** (1.135) | 5.575 *** (1.681) | 17.57 *** (6.815) | |
| Observations | 1207 | 1207 | 1175 | 1173 |

Notes: Standard errors are in parentheses. ***, **, and * indicate significance at 1%, 5% and 10% respectively. Insignificant coefficients are excluded from the table of results.

Pre-existing rights to patent holdings (*d_PAT_FILED*) increase the probability that innovation will occur, in both manufacturing and service firms.[9] These findings support the work of Aghion et al. (2015)

---

[9] This study uses patent data as an innovation input. Frequently though, patent data is used as a measure of innovation output. However, its use as a measure of output is not unproblematic. Patents measure inventions rather than innovations (Coombs et al. 1996; Flor and Oltra 2004; Organisation for Economic Co-Operation and Development 2005). Measuring innovation using patent data risks overestimating the level of innovativeness by counting inventions that have not been transformed into marketable innovations (Becheikh et al. 2006). Further, many innovations are not patented, and some are covered by multiple patents (Organisation for Economic Co-Operation and Development 2005). For several reasons (for example, high costs or difficulties in patenting process), some firms prefer to protect their innovations by other methods

in relation to the strength of patent rights for the innovation process or product. For service firms, a number of additional factors explain their likelihood of innovating. Two are measures of firm size. The first, number of workers employed (*lnLABOR11*) has a positive impact, suggesting that firms with larger staff complements are more likely to innovate in the future. This is indicative of economies of scale, which facilitates the firm's ability to have very specialist functions for employees, for example, a department focused on innovation. The second, value of firms' domestic sales (*lnDSALES*) has a negative impact, implying that higher sales revenues by service firms reduce their odds of innovating; in other words, a good performance disincentivizes innovation. There is evidence of a difference between domestic service firms and those with foreign ownership. Firms with a level of foreign ownership (*d_HFDI*) are more likely to innovate, perhaps indicative of a higher degree of risk-taking on the part of foreign owners and recognition of the benefits from innovation.

Regarding innovation intensity, there are two common determinants for manufacturing and service firms, domestic sales (*lnDSALES*) and collaborating for innovation purposes (*d_O_STRATEGY*). Unlike its impact on the probability of innovating, larger sales revenues (*lnDSALES*) are a driver of innovation intensity, unsurprising, as innovation is potentially costly and risky. So while greater revenues are associated a lower probability of innovation, it appears that after a decision to innovate is made, greater sales fuels investment in innovative activities. In contrast, collaborating (*d_O_STRATEGY*) reduces innovation intensity by each firm, as it spreads the inherent risks associated with such activity.

Other significant determinants of innovation intensity for service firms are revenues from exports (*lnESALES*), the size of the workforce (*lnLABOR11*), and technological capacity (*d_TECH_BASED*). Higher levels of exports are associated with greater intensity, reinforcing the influence of domestic sales (*lnDSALES*). This could be due to competition that exporting firms face in the international marketplace, which compels them to innovate to remain competitive. The positive relationship between technological inputs and innovation intensity indicates that technology is an important factor of innovation in services, allowing for greater efficiency and effectiveness. Larger services firms have lower innovation intensity than smaller firms; specifically, a higher complement of employees is associated with lower average investment in innovation, suggesting the existence of scale economies in innovation intensity in the services sector.

## 4.2. Innovation Output

Next, the determinants of innovation output are examined (column 4 of Tables 4 and 5). For manufacturing and service firms, such output is negatively associated with the *predicted* level of innovation intensity (*lnINEM_hat*), which lies in contrast to findings from previous research (see Crespi and Zúñiga 2012; Crespi et al. 2014; Masso and Vahter 2011). A possible explanation is inefficient use of investment funds for innovation purposes. The availability of technology and technical information (*d_TECH_INFO_OBS*) and information on new market trends (*d_MK_INFO_OBS*) are major obstacles to firms in both sectors producing innovative products and processes. Additional obstacles for service firms are organizational and managerial self-confidence in their ability to successfully innovate (*d_FCULTURE_OBS*) and the level of compliance requirements to meet international standards (*d_FRM_OPEN_OBS*).

Innovation output is positively related to the level of domestic sales (*lnDSALES*) for manufacturing and service firms and foreign sales (*lnESALES*) in the case of services firms only, and firms' technological capacity (*d_TECH_BASED*). The size of firms' workforce (*lnLABOR12*) is significant but varies in its effect on the two sectors. In manufacturing, firms with a larger number of workers have

---

such as maintaining lead time over rivals, industrial secrecy, and technological complexity (Archibugi and Pianta 1996; Mansfield 1985; Kleinknecht et al. 2002). Since not all innovations are patentable, and not all patentable innovations are patented (Dulude 1985), patent data is thus a very imprecise measure of innovation output (Becheikh et al. 2006). For a thorough survey of the problems with the use of patents to measure innovation activity, see Desrochers (1998).

greater innovation output. For service firms, the converse is true; firms with more employees have smaller output. This implies that in the service sector larger service firms are less efficient than smaller firms in converting innovation inputs to innovation outputs.

*4.3. Innovation Productivity*

The impact of innovation on average productivity growth is shown in column 5 of Tables 4 and 5. The results indicate that higher innovation intensity (*lnINEM_hat*) and innovation output (*d_INNOV*) are associated with greater productivity growth in both sectors.

In the manufacturing sector, productivity growth is higher when firms are larger (*lnLABOR11*), there is significant foreign ownership (*d_HFDI*), and the younger the firm (*lnFIRM_AGE13*). Additionally, the cost of finance (*d_FINANCE_COST*) and obtaining business licenses and permits (*d_BUS_LICENCE*)—normally obstacles—enhance productivity in manufacturing firms, implying a high degree of business facilitation in the financial and public sectors. For the services sector, productivity growth is also higher when firms are larger (*lnLABOR12*), but older (*lnFIRM_AGE13*), in contrast to manufacturing firms. Further, the reliability of electricity supply (*d_ELECTRICITY*), tax rates (*d_TAX*) and crime (*d_CRIME*) increase productivity, implying that these barriers are low. Another factor is technology (*d_TECH_BASED*); firms with greater technological know-how have lower productivity growth.

## 5. Conclusions

Caribbean countries have experienced low growth rates for over a decade. It is recognized that the private sector is critical in reversing this trend. However, the sector has not performed in a manner consistent with this objective. A key channel through which the private sector can improve its performance is innovation. Innovation is a major source of growth and a necessary condition for competitiveness. Little is known about the level of innovation in the Caribbean. Available evidence suggests that countries from the region, individually and collectively, lag behind their counterparts in other regions of the world. Given the dearth of studies on innovation in small states, this paper examined factors of innovation and innovation performance in the Caribbean at the firm level, with a special emphasis on the manufacturing and services sectors. The econometric analysis examines impacts on the innovation process at several different stages: the decision to innovate, impacts on innovation performance and on overall innovation productivity.

The paper employed data from the PROTEqIN survey, which covered 1928 firms from across the Caribbean. Results show that patent rights increase the likelihood of innovating in regional firms, while the level of domestic sales and the extent to which collaboration for innovation purposes takes place determine investment in innovation. The level of innovation intensity (more specifically, the efficiency with which innovation funds are managed), availability of technology, knowledge about new market trends, domestic sales and the size of the workforce determine innovation output. Innovation productivity is driven by innovation intensity, the level of innovation output, technological capacity, and the size of the workforce. Although the age of the firm is important for both manufacturing and service firms, the impact runs in different directions; younger manufacturing but older service firms innovate. Taken together these results indicate that smaller, older service firms, and larger, but younger manufacturing firms, on average engage in more innovation activities. This may possibly be due to the risk aversion of older manufacturing firms, which impedes the introduction of innovation in processes or organizational structures.

Apart from the similar factors of innovation from both sectors, several differences exist. Domestic sales, collaboration for innovation purposes, and foreign ownership are significant determinants in explaining the probability that service firms will innovate, in addition to those outlined previously. Innovation intensity in service firms is also driven by the level of exports, size of the workforce, and technological sophistication. Foreign ownership, compliance with international standards and export sales are other factors of innovation output for service firms. Organizational and managerial

confidence, electrical reliability, taxation and crime help to drive innovation productivity among service firms, while the cost of finance and obtaining business licenses and permits have a positive effect for manufacturing firms only. Therefore, international competition and nationality diversity in the ownership structure are characteristic of service firms that innovate. This may indicate that operating in international markets provides greater potential for innovation and/or that it may provide greater incentives to engage in novel innovation activities for service firms. Different barriers to innovation affect firms from each sector. The positive relationship with innovation productivity suggests that these barriers are low, and underline the importance of business facilitation to innovation efforts.

The differences between firms from each sector questions the validity of one-size-fits-all national schemes for incentivizing innovation. Results strongly suggest that policymakers must be sensitive to these differences when creating policy frameworks. On the firm side, performance will depend on a commitment to undertaking innovative activities, notwithstanding the lack of guaranteed positive outcomes.

**Author Contributions:** Antonio Alleyne conceptualized the original research, wrote the methods section, carried out the econometric estimations, and presented the results. Troy Lorde updated the original research concept, analyzed the econometric results, wrote the majority of the paper, and undertook revisions as recommended by the referees and editor. Quinn Weekes presented an earlier version of this paper, provided suggestions for technical improvement at each stage, and made contributions to the write-up of the paper.

**Conflicts of Interest:** The authors declare no conflict of interest.

## Appendix A

**Table A1.** Description of Model Variables.

| Variable | Description | Equation | | |
|---|---|---|---|---|
| | **Endogenous Variable** | Equation (1) & Equation (2) | Equation (4) | Equation (5) |
| *d_INF* | Firm reports any type of innovation activity. Does this establishment have a department or a group of professionals dedicated to innovation (research and development, service); In the last three years, did this establishment introduce to the market a new or significantly improved good or service; In the last three years, did this establishment introduce improvements in marketing of its goods or services? | √ | | |
| *lnINEM* | Natural logarithm of firm's total investment in innovation per average employee (between 2011 and 2012) | √ | √ | |
| *d_INNOV* | 1 if firm has introduced at least one product or process innovation in the last three years; 0 otherwise | | √ | √ |
| *lnPROD* | Increased firm's performance per employee expressed in natural logarithms: sales per employee in 2012 less sales per employee in 2011 | | | √ |
| | **Continuous Independent Variables** | | | |
| *lnFIRM_AGE13* | Natural log of firms age at 2013, the previous year to which the survey taken | | | |
| *lnLABOR11* | Natural log of total labor force in 2011 | √ | | |
| *lnLABOR12* | Natural log of total labor force in 2012 | | √ | √ |
| *lnINEM_hat* | Estimated value of expenditure on innovation per average employee (between 2011 and 2012) | | √ | √ |
| *INNOV_hat* | Estimated probability of introducing product or process innovation | | | √ |
| *lnDSALES* | Natural log of firm's share of domestic sales (less indirect exports) in total sales | √ | √ | |
| *lnESALES* | Natural log of firm's share of exports in total sales | √ | √ | |

**Table A1.** *Cont.*

| Variable | Description | Equation | | |
|---|---|---|---|---|
| | **Endogenous Variable** | Equation (1) & Equation (2) | Equation (4) | Equation (5) |
| **Binary Indicator Variables** | | | | |
| *d_HFDI* | 1 if firm reports more than 10% foreign ownership | √ | √ | |
| *d_PAT_FILED* | 1 if firm filed any patent application during the last 3 years (2011–2013); 0 otherwise | √ | | |
| *d_PUB_FUND_U* | 1 if firm accesses any type of public fund for innovation; 0 otherwise | √ | | |
| *d_O_STRATEGY* | 1 if firm reports any type of collaboration for innovation; 0 otherwise | √ | | |
| *d_MKT_COMP* | 1 if firm considers market competition very important/critical innovation; 0 otherwise | √ | | |
| *d_TECH_BASED* | 1 if firm is technologically driven entity, determined by the technology recruitment requirements of staff; 0 otherwise | √ | | √ |
| **Categorical Variables** | | | | |
| OPTIONAL RESPONSES: No Obstacle (1); Minor Obstacle (2); Moderate Obstacle (3); Major Obstacle (4); Very Severe Obstacle (5) | | | | |
| *d_FCULTURE_OBS* | Current organizational/managerial degree of self-confidence for innovation | | | √ |
| *d_FINANCE_OBS* | Level of available financial resources | | | √ |
| *d_EMP_QTY_OBS* | Qualification of employees | | | √ |
| *d_PPOLICY_OBS* | Copy right and patent protection against copycats | | | √ |
| *d_MK_INFO_OBS* | Level of information on new trends of the market | | √ | √ |
| *d_TECH_INFO_OBS* | Technical uncertainties; Level of information on available technologies | | √ | √ |
| *d_FRM_OPEN_OBS* | Compliance requirements to international standards | | | √ |
| *d_OTHER_OBS* | Other specified conditions reported as obstacles | | √ | |
| OPTIONAL RESPONSES: No Obstacle (1); Minor Obstacle (2); Moderate Obstacle (3); Major Obstacle (4); Very Severe Obstacle (5) | | | | |
| *d_ELECTRICITY;* | Electricity level can affect the current operations | | | √ |
| *d_TAX;* | Tax rates can affect the current operations | | | √ |
| *d_CRIME;* | Crime, theft and disorder levels can affect the current operations | | | √ |
| *d_FINANCE_COST;* | Cost of finance (e.g., interest rates) can affect the current operations | | | √ |
| *d_BUS_LICENCE* | Business licensing and permits can affect the current operations | | | √ |

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
