# Peer review of "A Firm-Level Investigation of Innovation in the Caribbean: A Comparison of Manufacturing and Service Firms"

_economies, doi:10.3390/economies5030034_

Round 1

Reviewer 1 Report

This paper aims to address an important topic that has been well documented elsewhere but is underexamined in the context of Caribbean countries. This intended objective is not successfully achieved by the manuscript in its current form. A major problem with the paper is a lack of theoretical/analytical framework. It is well known that innovation is a complex process influenced and determined by a multitude of factors such as firm-level organizational factors and regional/national-level institutional factors etc. And it goes without saying that the importance of different determinant factors varies across different regions. In the case of Caribbean countries, what is the main factor leading to the poor innovative performance in this region? The author(s) lists out a number of independent variables that may affect the innovative incentives and performance of business firms in Caribbean countries. But no where is any justification provided to explain on what theoretical/conceptual basis are these indicators included? The paper does not highlight the key hypothesis that will be tested in the Caribbean countries. Simply listing out many variables that are associated with firm innovativeness does not contribute any new knowledge/insight to the traditional topic for which numerous articles have been produced. 

Below are some detailed comments:

(1) The section of literature review should be substantially rewritten to critically evaluate the existing findings on determinants of innovation in the mainstream literature in economics and social sciences at large. This part should also explain the reason for the case study of Caribbean countries. What are the hypothesized differences in the pattern and process of innovation between Caribbean countries and elsewhere? Why we have to bother to make a distinction between manufacturing industry and service industry?

(2) The validity for the selection of some variables is dubious. For example, the author(s) uses d_PAT_FILED as independent variable to evaluate its effect on the probability of innovation. However, d_PAT_FILED is defined as whether or not firm filed any patent application during the last 3 years. It is self-evidence that firms which filed any patent application in the past years were more likely to conduct innovative activities. Moreover, the author(s) uses the absolute value of domestic/export sales to examine their effects on innovative performance. But a commonly utilized indicator in innovation studies is to calculate the share of exports in total output value in order to measure the impact of degree of export exposure/dependence. 

(3) In line 305-306, the author(s) states that firms with larger staff are more likely to innovate. The implication of this findings is quite ambiguous. Does it represent the presence of economies of scale or the intensity of R&D input? The share of R&D staff in total staff should be included in order to elucidate the nature of this statistical result.

(4)line 332-226. The paper identified an intriguing pattern that innovative output is negatively associated with innovation intensity. The author(s) argues that this could be because firms with innovative incentives are more likely to encounter barriers in the innovative process. This does not make sense and lacks empirical support. 

(5) Line 348, the paper states that for service firms, firms with more employees are less innovative. But in line 385, the paper says larger service firms are more likely to engage in innovation activities. These self-contradictory conclusions should be reconciled through careful checking. 

(6) line 382-382, The paper reveals that the impact of age variable runs in different directions in manufacturing and service industries. This is an interesting finding. But the paper should go further to explain why this happens in the context of Caribbean countries.  

Author Response

This paper aims to address an important topic that has been well documented elsewhere but is underexamined in the context of Caribbean countries. This intended objective is not successfully achieved by the manuscript in its current form. A major problem with the paper is a lack of theoretical/analytical framework. It is well known that innovation is a complex process influenced and determined by a multitude of factors such as firm-level organizational factors and regional/national-level institutional factors etc. And it goes without saying that the importance of different determinant factors varies across different regions. In the case of Caribbean countries, what is the main factor leading to the poor innovative performance in this region? The author(s) lists out a number of independent variables that may affect the innovative incentives and performance of business firms in Caribbean countries. But no where is any justification provided to explain on what theoretical/conceptual basis are these indicators included? The paper does not highlight the key hypothesis that will be tested in the Caribbean countries. Simply listing out many variables that are associated with firm innovativeness does not contribute any new knowledge/insight to the traditional topic for which numerous articles have been produced. 

Thank you for your comments.  We have pointed in the literature review to the fact that the work on innovation can be traced to Schumpeter’s seminal works.  From there we go on discuss various determinants of innovation.  We cannot speak with any conviction on the reasons for the poor innovative performance in the Caribbean, because the data record is poor as we have alluded to.  We identify possible reasons for this state of affairs in lines 81-89.  In Section 3.2, we state that we adopt the variables employed by previous authors.  These studies were for countries with whom the Caribbean is typically grouped, i.e., Latin America.

Below are some detailed comments:

(1) The section of literature review should be substantially rewritten to critically evaluate the existing findings on determinants of innovation in the mainstream literature in economics and social sciences at large. This part should also explain the reason for the case study of Caribbean countries. What are the hypothesized differences in the pattern and process of innovation between Caribbean countries and elsewhere? Why we have to bother to make a distinction between manufacturing industry and service industry?

Thank you for your comments.  The literature review was enhanced to include existing findings on the determinants of innovation in economics and social sciences in lines 127-167.  The study starts from a position of ignorance and moves towards trying to identify possible determinants of innovation in the Caribbean based on the data found in the data set we have. The revised paper provides arguments for distinguishing between services and manufacturing in lines 160-183.

(2) The validity for the selection of some variables is dubious. For example, the author(s) uses d_PAT_FILED as independent variable to evaluate its effect on the probability of innovation. However, d_PAT_FILED is defined as whether or not firm filed any patent application during the last 3 years. It is self-evidence that firms which filed any patent application in the past years were more likely to conduct innovative activities. Moreover, the author(s) uses the absolute value of domestic/export sales to examine their effects on innovative performance. But a commonly utilized indicator in innovation studies is to calculate the share of exports in total output value in order to measure the impact of degree of export exposure/dependence. 

Thank you for your comments.  The domestic and export sales data used in the study are in actuality the shares of each in total sales.  The table describing the variables in question has been amended to make this clearer.  Filing a patent increases the likelihood of innovating, but “how likely” is it to lead to innovating activities is still an open question.  This is what we were trying to determine.

(3) In line 305-306, the author(s) states that firms with larger staff are more likely to innovate. The implication of this findings is quite ambiguous. Does it represent the presence of economies of scale or the intensity of R&D input? The share of R&D staff in total staff should be included in order to elucidate the nature of this statistical result.

Thank you for your comments.  We have included your suggestion as an explanation for this finding, which appears in lines 328-330 in the revised manuscript.  Unfortunately, the data set does not include information on the number of R&D staff in absolute or relative terms.

(4)line 332-226. The paper identified an intriguing pattern that innovative output is negatively associated with innovation intensity. The author(s) argues that this could be because firms with innovative incentives are more likely to encounter barriers in the innovative process. This does not make sense and lacks empirical support. 

Thank you for your comments.   The manuscript actually said that firms with a greater focus on investment in innovation, rather than firms with innovative incentives, are more likely to encounter barriers in the innovative process.  We have suggested that the finding may be due to inefficiently spend of innovation funds (line 360 in revised manuscript).

(5) Line 348, the paper states that for service firms, firms with more employees are less innovative. But in line 385, the paper says larger service firms are more likely to engage in innovation activities. These self-contradictory conclusions should be reconciled through careful checking. 

Thank you for pointing out the inconsistency.  The correction has been made (lines 408-409 in revised manuscript).

(6) line 382-382, The paper reveals that the impact of age variable runs in different directions in manufacturing and service industries. This is an interesting finding. But the paper should go further to explain why this happens in the context of Caribbean countries.  

Thank you for the suggestion.  We have offered an explanation in lines 409-410.

Reviewer 2 Report

The paper assess empirically the determinants of innovation on a sample of Caribbean countries, finding differences between manufacturing and service firms. The structure and findings of the paper are relevant and important, given the scarce number of papers focused on that topics in developed countries, and particularly in The Caribbean. 

I have the following comments and recommendations:

i. The review of the literature should include a brief discussion on the definitions and differences between products (manufacturing) and services. I suspect that authors use ISIC classification in their methodology but no specific reference is made to a qualitative/conceptual differences among services vs manufacturing activities.

ii.  Notation in section 3.1. should be clariffied. For example, Qi for equation (1) is defined in lines 219-221; but another definition of Qi is introduced in lines 224-225 for equation (2). If variables are different, different notations should be used. Otherwise the relationship between both varaibles should be explicited. The same can be said on the variable Inf i. In equation (1) is defined as binary variable while in equation (2) is defined as a quantitative variable, firm's investment and R&D pero employee. 

iii. Terms of expression (3) between lines 229-230 should be defined in order to understand clearly the empirical strategy used by authors.

iv. Section 3, table 3 presents characteristics of the sample. It should be interesting to include the share of firms considered according their country of origin (otherwise justify the reason for not including it). 

v. It is reasonable to suspect that the quality of public goods provided by states, influence on the performance of firms (that explains the differences between countries described in section 1 in international rankings). So it would be important to consider the country of origin -may be grouped by size of country or size of government (among others)-, in order to inquire on the effect of this variable on innovation of firms. 

Author Response

The paper assess empirically the determinants of innovation on a sample of Caribbean countries, finding differences between manufacturing and service firms. The structure and findings of the paper are relevant and important, given the scarce number of papers focused on that topics in developed countries, and particularly in The Caribbean. 

I have the following comments and recommendations:

i. The review of the literature should include a brief discussion on the definitions and differences between products (manufacturing) and services. I suspect that authors use ISIC classification in their methodology but no specific reference is made to a qualitative/conceptual differences among services vs manufacturing activities.

Thank you for your comments.  We have included a brief discussion in the literature review on the differences between products and services in lines 168-183.

ii.  Notation in section 3.1. should be clariffied. For example, Qi for equation (1) is defined in lines 219-221; but another definition of Qi is introduced in lines 224-225 for equation (2). If variables are different, different notations should be used. Otherwise the relationship between both varaibles should be explicited. The same can be said on the variable Inf i. In equation (1) is defined as binary variable while in equation (2) is defined as a quantitative variable, firm's investment and R&D pero employee. 

Thank you for pointing out the lack of clarity in our model details.  We have used different variable names to account for the differences between the Qi’s in the original manuscript. The “second” Qi is now referred to as .  An adjustment was also made to clarify the differences between infi in both equations.

iii. Terms of expression (3) between lines 229-230 should be defined in order to understand clearly the empirical strategy used by authors.

iv. Section 3, table 3 presents characteristics of the sample. It should be interesting to include the share of firms considered according their country of origin (otherwise justify the reason for not including it). 

Thank you for this useful comment.  The information recommended was included in the revised Table 2.

v. It is reasonable to suspect that the quality of public goods provided by states, influence on the performance of firms (that explains the differences between countries described in section 1 in international rankings). So it would be important to consider the country of origin -may be grouped by size of country or size of government (among others)-, in order to inquire on the effect of this variable on innovation of firms. 

Thank you for this comment.  The main purpose in Section 1 was to highlight the very limited information on innovation in the Caribbean. This comment is very useful and one we will definitely consider going forward in our research agenda on innovation in the Caribbean.

Round 2

Reviewer 1 Report

The author(s)'s efforts in revising the paper are commendable. But some questions raised in my earlier report are not adequately addressed. 

(1) the author(s) argues that d_PAT_FILED is a valid independent variable to measure innovative input. I do not agree. It is well understood in innovation studies that filing a patent is a commonly utilized indicator of innovative output. It is unjustifiable to use one indicator of innovative output to explain another indicator of innovative output. 

(2) the author(s) found a strange pattern that innovative output is negatively associated with innovation intensity. The validity of this finding is quite dubious. And the author(s)'s explanation that firms with innovative intensity are more likely to encounter barriers in the innovative process is also ungrounded. Even if we follow this hypothesis, the main issue here determining the level of innovation output is not innovative intensity per se, but rather the efficiency in the management of innovative funds. The author(s) should then revise the main findings in abstract and conclusion. 

(3) The author(s)'s explanation about the different patterns of innovation for manufacturing and service firms is not convincing (line 405-407). If it is due to the risk aversion of older firms, why older service firms are more innovative?  

Author Response

The author(s)'s efforts in revising the paper are commendable. But some questions raised in my earlier report are not adequately addressed.

(1) the author(s) argues that d_PAT_FILED is a valid independent variable to measure innovative input. I do not agree. It is well understood in innovation studies that filing a patent is a commonly utilized indicator of innovative output. It is unjustifiable to use one indicator of innovative output to explain another indicator of innovative output.

Thank you for your comment.  We have addressed this concern by including a footnote in the revised paper, footnote number 9 which is restated here:

This study uses patent data as an innovation input.  Frequently though, patent data is used as a measure of innovation output.  However, its use as a measure of output is not unproblematic.  Patents measure inventions rather than innovations (Coombs, Narandren, & Richards, 1996; Flor & Oltra, 2004; OECD, 2005).  Measuring innovation using patent data risks overestimating the level of innovativeness by counting inventions that have not been transformed into marketable innovations (Becheikh, Landry, & Amara, 2006).  Further, many innovations are not patented, and some are covered by multiple patents (OECD, 2005).  For several reasons (for example, high costs or difficulties in patenting process), some firms prefer to protect their innovations by other methods such as maintaining lead time over rivals, industrial secrecy, and technological complexity (Archibugi, & Pianta, 1996; Mansfield, 1985; Kleinknecht, van Montfort, & Brouwer, 2002).  Since not all innovations are patentable, and not all patentable innovations are patented (Dulude, 1985), patent data is thus a very imprecise measure of innovation output (Becheikh, Landry, & Amara, 2006).  For a thorough survey of the problems with the use of patents to measure innovation activity, see Desrochers (1998).

We hope this can alleviate your concern.

(2) the author(s) found a strange pattern that innovative output is negatively associated with innovation intensity. The validity of this finding is quite dubious. And the author(s)'s explanation that firms with innovative intensity are more likely to encounter barriers in the innovative process is also ungrounded. Even if we follow this hypothesis, the main issue here determining the level of innovation output is not innovative intensity per se, but rather the efficiency in the management of innovative funds. The author(s) should then revise the main findings in abstract and conclusion.

Thank you for this comment.  What you are referring is from the original manuscript.  In the first revision of the paper, our explanation is precisely in line with your recommendation.  This can be found on lines 356-357 in both the first revision and the current revision. In the current revision, we added a phrase to the abstract and conclusion to allude more carefully to the efficiency in the management of innovative funds when the variable innovation intensity is referenced.

(3) The author(s)'s explanation about the different patterns of innovation for manufacturing and service firms is not convincing (line 405-407). If it is due to the risk aversion of older firms, why older service firms are more innovative? 

Thank you for your comment.  We realize that we were unclear when we said that “This may possibly be due to the risk aversion of older firms, which impedes the introduction of innovation in processes or organizational structures”.  What we meant was older manufacturing firms.  In the current revision, we added the word “manufacturing” to address the lack of clarity.

Round 3

Reviewer 1 Report

I accept the justifications provided by the author(s). I have no more questions on the latest revision.